# Image Denoising Using a Compressive Sensing Approach Based on Regularization Constraints

**DOI:** 10.3390/s22062199

**Published:** 2022-03-11

**Authors:** Assia El Mahdaoui, Abdeldjalil Ouahabi, Mohamed Said Moulay

**Affiliations:** 1AMNEDP Laboratory, Department of Analysis, University of Sciences and Technology Houari Boumediene, Algiers 16111, Algeria; aelmahdaoui@usthb.dz (A.E.M.); mmoulaydz@hotmail.com (M.S.M.); 2UMR 1253, iBrain, INSERM, Université de Tours, 37000 Tours, France

**Keywords:** compressive sensing, image reconstruction, regularization, total variation, augmented Lagrangian, nonlocal self-similarity, wavelet denoising

## Abstract

In remote sensing applications and medical imaging, one of the key points is the acquisition, real-time preprocessing and storage of information. Due to the large amount of information present in the form of images or videos, compression of these data is necessary. Compressed sensing is an efficient technique to meet this challenge. It consists in acquiring a signal, assuming that it can have a sparse representation, by using a minimum number of nonadaptive linear measurements. After this compressed sensing process, a reconstruction of the original signal must be performed at the receiver. Reconstruction techniques are often unable to preserve the texture of the image and tend to smooth out its details. To overcome this problem, we propose, in this work, a compressed sensing reconstruction method that combines the total variation regularization and the non-local self-similarity constraint. The optimization of this method is performed by using an augmented Lagrangian that avoids the difficult problem of nonlinearity and nondifferentiability of the regularization terms. The proposed algorithm, called denoising-compressed sensing by regularization (DCSR) terms, will not only perform image reconstruction but also denoising. To evaluate the performance of the proposed algorithm, we compare its performance with state-of-the-art methods, such as Nesterov’s algorithm, group-based sparse representation and wavelet-based methods, in terms of denoising and preservation of edges, texture and image details, as well as from the point of view of computational complexity. Our approach permits a gain up to 25% in terms of denoising efficiency and visual quality using two metrics: peak signal-to-noise ratio (PSNR) and structural similarity (SSIM).

## 1. Introduction

Compressed sensing (CS) has already attracted great interest in various fields. Examples include medical imaging [1,2], communication systems [3,4,5,6], remote sensing [7], reconstruction algorithm design [8], image storage in databases [9], etc. Compressed sensing provides an alternative approach to Shannon’s vision to reduce the number of samples and/or reduce transmission/storage costs. Other approaches also address this issue, such as random sampling [10]. Compressed sensing recovery is a linear optimization problem. The most common CS retrieval algorithms explore the prior knowledge that a natural image is sparse in certain domains, such as the wavelet domain, where simple and efficient noise reduction is possible [11,12,13,14,15], or in the discrete-gradient domain, which we will develop in this work.

In real life, images are usually noisy, but noise is random and is, therefore, unknown. This noise can have many origins: It can be due to poor weather conditions (wind, haze, fog, mist, …), light fluctuations, the electronic image sensor of a digital camera, the conditions under which the image was acquired or simply the manner the image was stored and the techniques used to compress it. Therefore, denoising is an essential preprocessing step to recover improved image quality.

We can consider the problem of image recovery as an inverse problem because the goal is to reconstruct an image close to the original image from a degraded image while respecting two constraints: an optimized image quality and the speed of execution. Our strategy is part of this framework, which consists in recovering a quality image—a debatable notion that is imprecise and depends not only on objective criteria but also on the “eye” of the observer—in a relatively short time.

The regularization of an inverse problem corresponds to the idea that data alone do not allow obtaining an acceptable solution and that it is, thus, necessary to introduce a priori information on the regularity of the image to be estimated, reconstructed or recovered. The regularization of an inverse problem corresponds to the idea that data alone cannot make obtaining an acceptable solution possible and that it is, therefore, necessary to introduce a priori information, likely allowing an estimation, reconstruction or recovery of the image of interest.

Many inverse problem optimization approaches for image denoising have been proposed in the literature. Some are based on deep learning and, more precisely, on the deep generative network [16], and others are based on models [17].

In particular, the total variation (TV)-based approach has been one of the most popular and successfully applied approach, where, for example, Chambolle [18] introduced the dual approach to the unconstrained real case. Subsequently, Beck and Teboulle [19] presented a fast computational method based on a gradient optimization approach to solve the TV-regularized problem. Recently, other denoising approaches have been proposed, such as a remote sensing image denoising method via a low-rank tensor approximation [20], which can be formulated as a generative Bayesian model. Another method for regularizing nonuniformly sampled data based on least-squares spectral analysis [21] can be applied to nonstationary signals by introducing classical sliding windowing, as was the case for processing marine seismic data. Other approaches to denoising hyperspectral images [22,23] combine total variation regularization and low-rank tensor decomposition.

However, since the total variation model favors piecewise constant image structures, the total variation models tend to oversmooth image details; it tends to smooth out the fine details of an image. To overcome these intrinsic drawbacks of the total variation model, we introduce a nonlocal self-similarity constraint as a complementary regularization. Nonlocal self-similarity can restore high-quality images. To make our algorithm robust, an augmented Lagrangian method was used efficiently to solve the above inverse problem.

The remainder of this paper is organized as follows. In Section 2, the basics of the methods used to image restoration are presented. The tools of denoising image considered in the proposed framework are described in Section 3. Section 4 presents our proposed algorithm, called DCSR. The experimental results from the processing of Lena, Barbara and Cameraman images, available in the public and royalty-free database https://ccia.ugr.es/cvg/dbimagenes/index.php (accessed on 1 June 2021), and the comparison with competing methods are shown in Section 5. The results are discussed in Section 6. Finally, conclusions are provided in Section 7.

## 2. Related Works

### 2.1. Compressive Sensing

Compressive sensing or compressed sensing [24,25] allows acquiring and efficiently reconstructing a signal by solving underdetermined linear systems. This technique is based on the principle where, by optimization, the sparsity of a signal can be exploited to recover it from several samples that are much lower than that required by the Nyquist-Shannon condition, which was, until recently, unavoidable.

Compressive sensing is based on three conditions: signal sparsity, the measurement matrix’s incoherence and the reconstruction algorithm’s robustness.

Let us assume a real signal x∈ℝN possessing a sparse representation in some transform domain, with K sparsity, K≪N. Then, the signal can be written as follows:(1)x=ψ u 
where ψ=ψ1,ψ2,…,ψN is the basis matrix, and u∈ℝN×1 is a weighted N dimensional vector with the following being the case:(2)  ui=〈xi, ψi〉=ψTx 
using ϕ as a sensing matrix of dimension M×N with K<M≪N. The measurement vector is acquired according to the following linear model:(3)f=ϕ x=A u 
with $A=\phi \;\psi^{-1}$. A is an M×N matrix that verifies the restricted isometry condition of order K.
(4)$$1-\delta_{K}\leq \frac{\|\;Au\;{\|}_{2}^{2}}{\|\;u\;{\|}_{2}^{2}}\leq 1+\delta_{K},\;0<\delta_{K}<1$$

The compressed sensing recovery of x from f can be obtained via $l_{1}$ norm minimization as the following optimization problem [26].
(5)$$\hat{u}=\min\|\;u\;{\|}_{1}\;\mathrm{subject}\;\mathrm{to}\;f=A\;u\;$$

More often, when f is contaminated by noise, the equality constraint is as follows:(6)$$\hat{\;u}=\min\|u{\|}_{1}\;\mathrm{subject}\;\mathrm{to}\|Au-f{\|}_{2}^{2}\;\leq \varepsilon$$
where ε>0 is the noise level. For an appropriate scalar weight α, we can obtain the following variant of (6).
(7)$$\underset{u}{\min\;}\frac{1}{2}\|Au-f{\|}_{2}^{2}+\alpha \;\|u{\|}_{1}$$

Problems (6) and (7) are equivalent because solving one will determine the parameter in the other such that both provide the same solution. The signal x can be reconstructed by solving the $l_{1}$ norm minimization problem under the condition that x is sufficiently sparse, and the measurement matrix ψ is inconsistent with the orthogonal basis ϕ.

Several iterative reconstruction algorithms solve the problem of denoising algorithms for CS recovery and obtained high performances for nature images, such as orthogonal matching pursuit [27] and approximate message passing (AMP) [28], an extension of AMP and denoising based AMP [29].

### 2.2. Augmented Lagrangian

The augmented Lagrangian method, originally known as the multipliers method [30], combines the Lagrangian function and a quadratic penalty term. It is applied to solve a constrained optimization problem iteratively:(8)minufu subject to H u=g 
where u∈ℝNand g∈ℝM, and H is a matrix of dimension M×N.

Applied to Equation (8), the augmented Lagrangian can be expressed as follows:(9) Lu,λ,ρ=f u− λTHu−g+ρ2‖Hu−g‖22
where λ∈ℝM is a vector of the Lagrange multiplier, and ρ>0 is the augmented Lagrangian penalty parameter for the quadratic infeasibility term [31]. We recall in Algorithm 1 the so-called Augmented Lagrangian method: this consists of searching for the optimal solution providing signal u by alternately updating u and the Lagrange multiplier λ. The alternate iteration method keeps one vector fixed and updates the other successfully until the stopping criterion is satisfied.
**Algorithm 1:** Augmented Lagrangian method.**Fixed:**k=0,  ρ>0,  λ0,  u0**Iterations:** for
k=k+1 until the stopping criterion is 

satisfied, repeat steps 1–2
1. Keep *λ* fixed and update *u*:
$$\;\;\;\;\;\;\;\;u_{k+1}=\underset{u}{\;\min\;}f\left(u\right)-\lambda_{k}^{T}\left(Hu-g\right)+\frac{\rho }{2}\|Hu-g{\|}_{2}^{2}$$2. Keep *u* fixed and update *λ*:
$$\;\;\;\;\;\;\;\;\;\;\lambda_{k+1}=\lambda_{k}+\rho \left(Hu_{k+1}-g\right)$$**Output:***u* the optimal solution.


## 3. Technical Framework for Image Denoising

Several methods exist for image denoising, including total variation image regularization [32], wavelet thresholding [33], nonlocal means [34], basis pursuit denoising [35], block matching and 3D filtering [36], among others. Moreover, these methods can perform, to a certain extent, image smoothing and preserve edges. Therefore, we can reconstruct the image well by CS theory to obtain more precise measurements of the original images than the corresponding noisy images.

In particular, we can insert them into two classes as exceptional cases of the proposed framework.

### 3.1. Regularization Functions

Image recovery in these application domains can be formulated as a linear inverse problem, which can be modelled as follows:(10)f=A u+ε
where f∈ℝM is the noisy image observation, u∈ℝN. is the cleanest image unknown, ε is an additive noise and A∈ℝM×N is a linear operator. Given A, image reconstruction extracts u^ from f, making classical least-squares approximation alone unsuitable. To stabilize recovery, regularization techniques are frequently used, producing a general reconstruction model of the following form:(11)arg minu12‖A u−f‖22+λ ϕregu
where λ>0 is a regularization parameter, and ‖ ‖2 denotes the l2 norm. The fidelity term, ‖ A u−f ‖22, forces the reconstructed image close to the original image, and the regularization function, ϕregu, performs noise reduction.

The choice of the regularization function is very important for reconstructing an image that reflects, as accurately as possible, the original image of interest. In our work, we have combined total variation and nonlocal self-similarities. The interest of such a choice is linked to the fact that the total variation (TV) model shows high efficiency in preserving the contours and recovering smooth regions. However, this operator is local. It, therefore, does not take into account nonlocal features of the data, such as repetitive structures (such as texture, for example). However, nonlocal self-similarity describes the repetitiveness of textures [37,38] or embodied structures in realistic images, which allows the preservation of sharp edges.

The total variation has been introduced first by Rudin, Osher and Fatemi [39] as a regularizing criterion for solving inverse problems. Then, the total variation model is written as follows:(12)TVu=‖Du‖p
where u∈ℝN represents an image, and D=Dh , Dv with Dh and Dv represents the gradient of the image in the horizontal and vertical direction, respectively. The lp norm could be the l1 norm corresponding to the anisotropic TV or the l2 norm corresponding to the isotropic TV. By definition, lp norm is ‖ u ‖p=∑ i=1 Nuip1p. In this paper, we consider p  to be equal to 1.

A nonlocal self-similarity (NLS) is a significant property of natural images too. It was proposed for the first time in image denoising [40] and was obtained in these steps. Image u  of size N  is divided into many overlapping blocks $u^{i}$ of size$\;\sqrt{n\;}\times \sqrt{n\;}$, at location i , i=1, 2,…, N. Then, find $\left(m-1\right)$ similar blocks, which comprise set $S_{u^{i}}$ in the training window with $\left(L\times L\right)$ size. Finally, all blocks of $S_{u^{i}}$ are stacked into a 3D array $Z_{u^{i}}$ of size $\sqrt{n\;}\times \sqrt{n\;}\times m$. Nonlocal self-similarity can be formulated as follows:(13)$$NLS\left(u\right)=\|\theta_{u}{\|}_{1}=\overset{N}{\underset{i=1}{\sum }}\|T^{3D}\left(Z_{u^{i}}\right){\|}_{1}$$
where $T^{3D}$ is a transform operator, and $\;T^{3D}\left(Z_{u^{i}}\right)$ includes the transform coefficients for $Z_{u^{i}}.\;\theta_{u}$ is the column vector of the lexicographically stacked representation of all 3D transform coefficients.

### 3.2. Wavelet Denoising

Wavelet shrinkage denoising removes whatever present noise and retains whatever signal is present regardless of the signal’s frequency content. In the wavelet domain, the energy of a natural signal is concentrated in a small number of coefficients; noise is, however, spread over the entire domain. The basic wavelet shrinkage denoising algorithm comprises three steps:
Discrete wavelet transform (DWT) [41];Denoising [42];Inverse DWT.


The following is the measurement model:(14) f=u+ε
where u is the original image of size M×N corrupted by additive noise ε. The goal is to estimate denoised image $\hat{u}$ from noisy observation f. The elimination of this additive noise permits the assumption that the appropriate decomposition basis allows the discrimination of a useful signal (image) from noise. This hypothesis justifies, in part, the traditional use of denoising by thresholding. Two thresholding methods are frequently used. The soft-thresholding function (also called the shrinkage function) is as follows.
(15)$$d_{j}^{\hat{u}}\left(k\right)=\left\{\begin{array}{cc} d_{j}^{y}\left(k\right)-S & \mathrm{if}\;\;d_{j}^{y}\left(k\right)>S\;\;\\ d_{j}^{y}\left(k\right)+S & \;\;\mathrm{if}\;\;d_{j}^{y}\left(k\right)<-S\;\\ 0 & \;\mathrm{otherwise} \end{array}\right.$$

The hard-thresholding function, another popular alternative, is as follows.
(16)$$d_{j}^{\hat{u}}\left(k\right)=\left\{\begin{array}{cc} d_{j}^{y}\left(k\right) & if\;\;\;\;\;\;\left|d_{j}^{y}\left(k\right)\right|>S\\ 0 & \;\;\;\;\;\;otherwise \end{array}\right.\;$$

$d_{j}^{y}\left(k\right)$ includes the wavelet coefficients of the measured signal at level j, and $d_{j}^{\hat{u}}\left(k\right)$ is the estimation of the wavelet coefficients of the useful signal with the threshold $S=\sigma \sqrt{2\log N}$, where N and σ represent the number of pixels for the test image and the standard noise deviation.

## 4. Our Efficient Image Denoising Scheme

In what follows, we substitute the aforementioned results (12) and (13) into (7), and we obtain the following problem.
(17)$$\underset{\;\;\;\;\;\;\;\;\;\;\;\;\;u}{\arg\;\min}\frac{1}{2}\|Au-f{\|}_{2}^{2}+\tau \;TV\left(u\right)+\mu \;NLS\left(u\right)$$

Let us recall that $TV\left(u\right)=\|Du{\|}_{1}$ and $NLS\left(u\right)=\|\theta_{u}{\|}_{1}=\sum_{1}^{N}\|T^{3D}\left(Z_{u^{i}}\right){\|}_{1}$.

Recovering image u with high quality requires solving the optimization problem described by Equation (17). This issue is converted into an equality-constraint problem by introducing w and x:(18)$$\underset{w,u,x}{\min}\frac{1}{2}\|Au-f{\|}_{2}^{2}+\tau \;\|w{\|}_{1}+\mu \;\|\theta_{x}{\|}_{1}\;\mathrm{subject}\;\mathrm{to}\;\;Du=w,\;u=x$$
where τ and μ are control parameters. The augmented Lagrangian function of (18) is as follows:(19)$$L_{A}\left(w,u,x\right)=\frac{1}{2}\|Au-f{\|}_{2}^{2}+\tau \;\|w{\|}_{1}+\mu \;\|\theta_{x}{\|}_{1}-\gamma^{T}\left(Du-w\right)-\varphi^{T}\left(u-x\right)+\frac{\mu }{2}\|Du-w{\|}_{2}^{2}+\frac{\beta }{2}\|u-x{\|}_{2}^{2}\;$$
where μ and β are the penalty parameters corresponding to$\;\|Du-w{\|}_{2}^{2}$ and $u-x_{2}^{2}$, respectively.

To solve (18), we use the augmented Lagrangian method iteratively as follows.
(20)$$\left\{\begin{array}{c} \left(w^{k+1},u^{k+1},x^{k+1}\right)=\underset{\;\;\;\;\;\;\;\;\;\;\;w,u,x}{\arg\;\min}\;\;L_{A}\left(w,u,x\right)\\ \begin{array}{c} \gamma^{k+1}=\gamma^{k}-\mu \left(Du^{k+1}-w^{k+1}\right)\\ \varphi^{k+1}=\varphi^{k}-\beta \left(u^{k+1}-x^{k+1}\right) \end{array} \end{array}\right.$$

Here, subscript k denotes the iteration index, and γ and φ are the Lagrangian multipliers associated with the constraints Du=w, u=x, respectively.

The alternative direction method is introduced to solve the problem efficiently. Due to the non-differentiability of Equation (19), the alternative direction method is introduced to solve the problem efficiently, which alternatively minimizes one variable while fixing the other variables to split Equation (19) into the following three subproblems. For the sake of simplicity, subscript k is omitted without confusion. Update w

Given u and x, we obtain subproblem w.
(21)$$\underset{\;\;\;\;\;\;\;\;\;\;w}{\arg\;\min}\;\tau \|w{\|}_{1}-\gamma^{T}\left(Du-w\right)+\frac{\mu }{2}\|Du-w{\|}_{2}^{2}$$

The optimization problem described by Equation (21) can be solved using the shrinkage formula [43]. Then, $\tilde{w}$ can be obtained as follows:(22)$$\tilde{w}=\max\left\{\left|Du-\frac{\gamma }{\mu }\right|-\frac{\tau }{\mu },\;0\right\}\mathrm{sgn}\left(Du-\frac{\gamma }{\mu }\right)$$
where $\max\;\left\{\cdot \right\}$ represents the larger number between two elements, and$\;\mathrm{sgn}\left(\cdot \right)\;$ is a piece- wise function that is defined as follows.
$$\mathrm{sgn}\left(x\right)=\left\{\begin{array}{cc} -1 & \mathrm{if}\;\;\;x<0\\ 0 & \mathrm{if}\;\;\;x=0\\ 1 & \mathrm{if}\;\;\;x>0 \end{array}\right.$$

Update u

By fixing w and x, the optimization associated with u is as follows.
(23)$$\underset{\;\;\;\;\;\;\;\;\;\;u}{\arg\;\min}\frac{1}{2}\|Au-f{\|}_{2}^{2}-\gamma^{T}\left(Du-w\right)-\varphi^{T}\left(u-x\right)+\frac{\mu }{2}\|Du-w{\|}_{2}^{2}+\frac{\beta }{2}\|u-x{\|}_{2}^{2}$$

Equation (23) is u-quadratic; thus, the minimization of the subproblem is simplified to a linear system.
(24)$$\left(D^{T}D+\left(\frac{\beta }{\mu }+\frac{1}{\mu }\right)I\right)\;\;u=\frac{1}{\mu }A^{T}\;f+D^{T}\left(w-\frac{\gamma }{\mu }\right)+\left(\frac{\beta }{\mu }x-\frac{\varphi }{\mu }\right)\;$$

The matrix on the left-hand side of the above system is positive, definite and tridiagonal, since $D^{T}D$ is a positive semidefinite tridiagonal matrix. Moreover, μ and β are both positive scalars.Update x

Given u, we obtain the x subproblem as follows.
(25)$$\underset{\;\;\;\;\;\;\;\;x}{\arg\;\min}\;\mu \;\|\theta_{x}{\|}_{1}+\frac{\beta }{2}\|u-x{\|}_{2}^{2}-\varphi^{T}\left(u-x\right)\;$$

By applying a completing square method and omitting all constants independent of x, the subproblem defined in Equation (25) can be simplified as follows.
(26)$$\underset{\;\;\;\;\;\;\;\;\;\;x}{\arg\;\min}\frac{1}{2}\|x-r{\|}_{2}^{2}+\frac{\mu }{\beta }\|\theta_{x}{\|}_{1}\;$$

Considering =u−c, with c=φβ, as a noisy observation of x, error (or noise) e=x−r follows a probability law that is not necessarily Gaussian, with zero mean and variance σ2. According to the central limit theorem or law of large numbers, the following equation holds:(27)1N‖x−r‖22=1K‖θx−θr‖22
with e, x, r ∈ ℝN and θx ,θr∈ ℝi for i=1,…,N.

Incorporating (27) into (26) results in the following.
(28)$$\underset{\;\;\;\;\;\;\;\;\;\;\theta_{x}}{\arg\;\min}\frac{1}{2}\|\theta_{x}-\theta_{r}{\|}_{2}^{2}+\frac{K\mu }{N\beta }\|\theta_{x}{\|}_{1}$$

Since $\theta_{x}$ is component-wise separable and an unknown variable, according to [44], the closed-form of the minimization problem of Equation (28) can be written as follows.
(29)$$\hat{\theta_{x}}=\mathrm{soft}\left(\theta_{r},\sqrt{2\varphi }\right),\;\;\varphi =\frac{K\mu }{N\beta }\;,\;\;K=\sqrt{n\;}\times \sqrt{n\;}\times m\;$$
(30)$$\hat{\theta_{x}}=\mathrm{sgn}\left(\theta_{r}\right)\max\left\{\left|\theta_{r}\right|-\sqrt{2\varphi ,\;}\;\;0\right\}$$

Thus, the solution of the x subproblem of Equation (25) is as follows:(31)$$\tilde{x}=\Omega \left(\hat{\theta_{x}}\right)$$
where Ω is the reconstruction operator.

Based on the discussions above, we obtain Algorithm 2 for solving (17):

Algorithm 2 is applied to recover corrupted images by white Gaussian noise and salt pepper noise.

The comparative performance of competing methods for the recovery of noisy images is discussed in the next sections.
**Algorithm 2:** Our algorithm (DCSR).**Input**: The measurement f and the linear measurement matrix A**Initialization:**
γ0=φ0=0, u0=f, w0=x0=0**while** Outer stopping criteria unsatisfied do**while** Inner stopping criteria unsatisfied douse Equation (22) to solve w sub-problem use Equation (24) to solve u sub-problem use Equation (31) to solve x sub-problem **end while**
Update multipliers by using Equation (20)**end while****Output:** The final image u is restored.

## 5. Experimental Results

In this section, we verify the efficiency and practicability of the proposed method by describing experiments with simulated and real data sets. The proposed algorithm’s performance, DCSR, is evaluated by comparing it with three other popular CS recovery algorithms: The first algorithm is wavelet thresholding, which involves denoising natural images by assuming that they are sparse in the wavelet domain. It transforms signals into a wavelet basis, thresholds the coefficients and then inverses the transform. The second algorithm extends Nesterov’s smoothing technique to TV minimization by modifying the smooth approximation of the objective function, called the NESTA algorithm. The third algorithm, group-based sparse representation (GSR), simultaneously enforces image sparsity and self-similarity under a unified framework in an adaptive group domain.

We used three images, Barbara and Cameraman gray-scale images with sizes of 256×256 and a color Lena image sized 512×512 in our experiments (see Figure 1).

The simulations used two types of noise: additive white Gaussian noise (AWGN) measured by its standard deviation σ and its mean m and the salt and pepper noise.

The salt and pepper noise in image has the following form:(32)ηp=vmax      with probability  q1  vmin     with probability  q2
where p=p1, p2,…,pn∈ℝn pixels, vmax is the pixel intensity of salt pixels and  vmin is the pixel intensity of pepper pixels. The sum, q=q1+q2, is the level of the salt and pepper noise.

### 5.1. Visual Quality Comparison

First, we evaluate the performance of our DCSR algorithm by performing experiments on Barbara’s image corrupted by white Gaussian noise with a standard deviation σ ranging from 20 to 80. Then, to verify the superiority of the proposed method, we compared it with the GSR algorithm, wavelet denoising and the NESTA algorithm. In Figure 2, each line represents the Barbara image reconstructed according to the following algorithms: DCSR (ours) in the first line, GSR in the second line, wavelet denoising in third line and NESTA in the fourth line. The level of the initial Gaussian white noise varies according to each column: (a) σ=20, (b) σ=50, (c) σ=60 and (d) σ=80.

The Barbara image is relatively complex given its rich texture and its geometric structure: it is clear that our algorithm (see the first line of Figure 2) has this ability to effectively denoise while preserving the details and texture of this particular image. Secondly, we handled the proposed approach to process impulsive salt and pepper noise in various noise levels varying from 20% to 70% and compared its denoising performance with several denoising algorithms: GSR algorithm, NESTA algorithm, and wavelet denoising. Figure 3 provides visual results for different algorithms and several noise levels; it can be observed that the Cameraman image has been reconstructed very well by our DCSR algorithm. By analyzing the images in the fifth column of Figure 3, obtained by the NESTA algorithm, we can observe some artifacts above the camera head and in the upper right corner due to the nature of the NESTA algorithm itself, which is characterized by a loss of precision in high-frequency components. We can conclude that the DCSR method presents the best quality image compared to other methods.

### 5.2. Image Quality Metrics

Peak signal-to-noise ratio and mean square error (MSE) have long been used as fidelity metrics in the image processing community. The formulas are simple to understand and implement; they are easy and fast to compute, and minimizing MSE is also very well understood from a mathematical point of view.

PSNR (peak signal-to-noise ratio, unit: dB) [45] is the ratio between the maximum possible power of a signal and the power of noise. Higher PSNR value means better visual quality. PSNR is defined as follows.
(33)$$\mathrm{PSNR}=10\;\log_{10}\frac{{\left(2^{k}-1\right)}^{2}}{\mathrm{MSE}}$$
(34)$$\mathrm{MSE}=\frac{1}{MN}\overset{M}{\underset{i=1}{\sum }}\overset{N}{\underset{j=1}{\sum }}{\left(u\left(i,j\right)-\hat{u}\left(i,j\right)\right)}^{2}$$

MSE is the mean square error between initial image u and estimated image $\hat{u}$ of size M×N; i and j represent the image row and column pixel position, respectively; and k is the number of bits of each sample value.

Structural similarity (SSIM) textures [46] have proven to be a better error metric for comparing the image quality with better structure preservation. They are in the range of [0,1], which is a value closer to one indicating better structure preservation:(35)$$\mathrm{SSIM}=l\left(u,\hat{u}\right)\times c\left(u,\hat{u}\right)\times s\left(u,\hat{u}\right)$$
such that $l\left(i,j\right)$ is the luminance comparison defined as follows:(36)$$l\left(i,j\right)=\frac{2\mu_{i}\mu_{j}+c_{1}}{\mu_{i}^{2}+\mu_{j}^{2}+c_{1}}$$
where $\mu_{i}$ and $\mu_{j}$ are functions of the mean intensities of signals i and j, respectively.

$c\left(i,j\right)$, the contrast comparison, is a function of standard deviations $\sigma_{i}$ and $\sigma_{j},$ and it is defined as the following form.
(37)$$c\left(i,j\right)=\frac{2\sigma_{i}\sigma_{j}+c_{2}}{\sigma_{i}^{2}+\sigma_{j}^{2}+c_{2}}$$

Structure comparison $s\left(i,j\right)$ is defined as follows:(38)$$s\left(i,j\right)=\frac{\sigma_{ij}+c_{3}}{\sigma_{i}\sigma_{j}+c_{3}}$$
where $\mu_{i}$ and $\mu_{j}$ are the mean value of images u and $\hat{u}$, respectively.$\;\sigma_{i}^{2}$ and $\sigma_{j}^{2}$ represent the variance of u and $\hat{u}$, respectively. $\sigma_{ij}$ is the covariance of images u and $\hat{u}$.$\;c_{1}$, and $c_{2}$ and $c_{3}$ are constant.

Specifically, it is possible to choose$\;c_{i}=K_{i}^{2}D^{2}$ with i=1 and $K_{i}\ll 1$ and D as the pixel values’ dynamic range.

### 5.3. Quantitative Assessment

In this subsection, we evaluate the quality of image reconstruction. We compare these methods quantitatively; the peak signal-to-noise ratio and structural similarity indices are calculated for images with different algorithms.

Remember that, at first, Barbara was contaminated by Gaussian white noise with different values of standard deviation σ and denoising with other algorithms. The results of the PSNR values by various algorithms are shown in Figure 4. Let us recall that a higher PSNR indicates superior image quality and good performance of the algorithm. The values of PSNR illustrate that DCSR yields a higher PSNR than the other methods.

Figure 5 presents the performance analysis of four denoising methods for grayscale images corrupted by additive white Gaussian noise. We can see that DCSR, our algorithm, exhibits the best performance with high PSNR (low noise level) and low PSNR (high noise level).

Furthermore, to evaluate the effect of denoising our algorithm, DCSR, the grayscale image is corrupted by salt and pepper noise with different levels. Table 1 shows the quantitative assessment results of PSNR and SSIM relative to our algorithm, DCSR, for several values of noise levels, and the results were obtained using GSR, wavelet and NESTA.

The best results are highlighted in bold type font. Table 1 validates that our method has superiority in image reconstruction compared with the three other methods.

The performance of our method is confirmed in Figure 6, which illustrates PSNR output variations concerning the input PSNR.

The next challenge is to apply this DCSR algorithm to color or multidimensional images. This is essential since most digital images used in the modern world are not grayscale but usually operate in either RGB or YCbCr color spaces. Both these colorspaces are three-dimensional.

We evaluate the proposed DCSR algorithm on the Lena color image because this test image is interesting from the point of view of the mixture of details, flat regions, shadow areas and texture. We mainly compareed our proposed method to wavelet denoising and the GSR algorithm.

As mentioned previously, two types of noise were used in these experiments: AWGN with a standard deviation σ and several salt and pepper noise levels.

Our first experiment is to add white Gaussian noise with different σ values, σ=20, 50, 60 and 80, to the test image (here Lena), thus generating noisy observations, and the challenge is to determine the most efficient denoising method in terms of PSNR.

The denoising results of this image with different algorithm are shown in Figure 7. From Figure 7, the proposed method achieves the highest scores of PSNR in all cases, which fully demonstrates that the denoising results by the proposed method are the best both objectively and visually.

Figure 8 shows the output results (after reconstruction) evaluated at different noise levels. DCSR algorithm provides the best denoising performance.

In the second experiment, we added salt and pepper noise with different noise levels at 10%, 15%, 20% and 30% to the Lena test image. Then, we applied our denoising algorithm to restore the noisy images and compared them with two other algorithms: wavelet denoising and GSR algorithm. Figure 9 shows that the DCSR algorithm provides better visual quality results. This performance is confirmed by Figure 10, which represents the variations of the output PSNR vs. the input PSNR. The robustness of our algorithm relative to the noise level is, thus, verified.

From these results of PSNR and SSIM, in all the cases, the proposed method achieves the highest scores, which fully demonstrates that the restoration results by the proposed method are the best both objectively and visually.

### 5.4. Algorithm Robustness

The robustness of the proposed algorithm will be confirmed in this subsection.

The test images are corrupted, on the one hand, by additive white Gaussian noise and, on the other hand, by salt and pepper noise at different noise levels. Figure 11, Figure 12, Figure 13 and Figure 14 show PSNR values (after reconstruction) as a function of the number of iterations for the greyscale images of Barbara and Cameraman and for the color image of Lena using different algorithms. Note that we did not use the NESTA algorithm in the case of the color images because NESTA is not suitable for color images.

### 5.5. Computational Complexity

In the following paragraph, we will estimate the computational complexity of the proposed DCSR algorithm. It is clear that the main complexity of the proposed algorithm comes from total variation TV and the high cost of the nonlocal self-similarities (NLS).

Knowing that the computational complexity of TV is $O\left(N\right)$ [47], let us compute that of NLS.

For an image u of N pixels, the average time to compute similar blocks for each reference block is$\;T_{s}$.

If $\sqrt{n\;}\times \sqrt{n\;}$ represents overlapped blocks $u^{i}$, i=1, 2,…, N and $\left(m-1\right)$ is the number of similar blocks denoted$\;S_{u^{i}}$, all blocks of $S_{u^{i}}$ are stacked in a matrix of size $\left(n\times m\right)$ with complexity$\;O\left(n\times m^{2}\right)$; hence, the resulting complexity is $O\left(N\;\left(n\;m^{2}+T_{s}\right)\right)$, similarly to the computational complexity of a group-based sparse representation GSR [48].

We point to the finding that the total computational complexity of our algorithm is $O\left(N\left(n\;m^{2}+T_{s}\right)+N\right)$.

It is interesting to compare the computational complexity of DCSR with competing methods.

The computational complexity of NESTA algorithm is $O\left(N+N\log_{2}N\right)$ [49] and that of wavelet denoising is $O\left(N\log_{2}N\right)$ [50].

Table 2 summarizes the computational complexity of the four algorithms used.

The result of this comparison is as follows.
(39)$$O\left(N\log_{2}N\right)<O\left(N+N\log_{2}N\right)<O\left(N\;\left(n\;m^{2}+T_{s}\right)\right)<O\left(N\left(n\;m^{2}+T_{s}\right)+N\right)\;$$

Relation (39) clearly shows that our proposed algorithm is more expensive in terms of computational complexity by an order N. Such an increase is not excessive in view of the good performance of this algorithm and the current computational means, which permit real-time processing.

## 6. Discussion

In the real world, the images acquired or measured or recorded have generally suffered degradation of various origins:-Bad weather conditions (wind, fog, haze, etc.);-Chain of acquisition of the image;-Compression of the image.

This degradation can be canceled or at least reduced by proceeding to a preprocessing procedure called a denoising operation or simply denoising. Such an operation allows, on the one hand, producing a perception of a quality image and, on the other hand, the improvement in the performance of subsequent image processing (extraction of the desired information, prediction, classification, texture analysis, segmentation, etc.).

With this in mind, the authors of reference [51] propose a new image denoising method called dehazing because it tends to eliminate haze due to bad weather conditions. This original method is based on the application of artificial multiexposure image fusion [52] involving local and global image details. Such an approach allows the recovery of quality images but, unfortunately, halos or artifacts often appear near the edges when the inputs are sparse, which makes postprocessing (linear saturation adjustment) introduced by the authors ineffective.

Wavelet image denoising [53] is powerful for edge detection in three preferred directions: diagonal, vertical and horizontal. For this purpose, several types of wavelets exist, such as the Haar wavelet that preserves edge information, but technically it is not continuous and is not differentiable. This wavelet can be identified with the optimization problem using an $l_{1}$ norm. In contrast, the Morlet wavelet [13], which is continuous, can be identified with the use of the $l_{2}$ norm.

Our method mixed the total variation with nonlocal self-similarities to recover the image without artifacts and preserve details and textures of the image. On a psycho-emotional level, the visual quality of an image is necessary for the observer. In this respect, it is interesting to note that the images reconstructed in Figure 2 by the NESTA algorithm are visually unpleasant and lose some important details. Limitations also appear during wavelet denoising: The noise contained in the image cannot be removed if the standard deviation is set too large (above 50), i.e., if the noise level is high. Concerning GSR (second row of Figure 2), we can observe that this algorithm is quite efficient in removing noise; however, it has a slight loss of contrast. On the other hand, it can be observed that our algorithm, DCSR, is efficient in removing even noise with standard deviations equal to 80. It is, therefore, confirmed that the proposed method provides the most visually satisfying results for both edges and textures.

On an objective and therefore measurable level, Table 1 shows that the PSNR and SSIM values obtained by NESTA are the lowest, which clearly indicates a limitation in the performance of this algorithm. GSR and wavelets obtain intermediate PSNR and SSIM values. At the end of the convergence analysis, Figure 11, Figure 12, Figure 13 and Figure 14 allow us to conclude that, as the number of iterations increases, all PSNR curves increase monotonically and stabilize from the 10th iteration. These figures show that all these algorithms converge very quickly. Nevertheless, the most robust algorithm should have a high PSNR. According to these figures, regardless of the nature of the noise or the test image used, DCSR has the highest PSNR. It is, therefore, the most robust algorithm among the competing methods.

However, when we compared our method, DCSR, in terms of computational complexity, our approach is not the most advantageous; however, real-time processing is largely feasible. Hence, complexity reduction techniques such as block-compressed sensing [54] or deep learning technique [55] are feasible.

We applied our algorithm in an ablation study to clarify the effect of total variation (TV) and nonlocal self-similarity (NLS) on the compressed sensing (CS) recovery model. First, we cancel all regularization constraints and leave only the CS alone. Next, we analyze CS coupled with TV and cancelled NLS. Then, we delete TV, replaced it with NLS and kept CS. Table 3 summarizes this ablation analysis using quantitative values: PSNR (dB) and SSIM. From Table 3, we can conclude that the presence of TV or NLS improves the quality of the reconstruction with CS, while noting that the CS+NLS coupling performs better than CS+TV. On the other hand, by comparing the four scenarios with each other, we can conclude that the reconstruction, with both regularization functions simultaneously (CS+TV+NLS), is significantly better than the other three scenarios: it effectively removes noise and improves the robustness of our approach.

## 7. Conclusions

In this paper, we proposed an original image denoising method based on compressed sensing that we called denoising-compressed sensing by regularizations terms (DCSR), by incorporating two regularization constraints in the model: total variation and nonlocal self-similarity. The optimization of this method is performed by the augmented Lagrangian, which avoids the difficult problem of nonlinearity and non-differentiability of the regularization terms.

The effectiveness of our approach was validated using images corrupted by white Gaussian noise and impulsive salt and pepper noise. Comparing DCSR in terms of PSNR and SSIM to state-of-the-art methods such as Nesterov’s algorithm, group-based sparse representation and wavelet-based methods, it turns out that, depending on the image texture and the type of noise corrupting the image, our method performs much better: We gain at least 25% in PSNR and at least 11% in SSIM. The price to pay is a slight increase in terms of computational complexity of the order of image size, but this does not call real time processing into question.

Due to the robustness and the speed of convergence of DCSR algorithm, its application is efficient in vital and sensitive domains such as medical imaging and remote sensing. Our future contribution is a technological breakthrough that includes introducing a layer of intelligence at the acquisition level aimed at automatically determining image texture and its quality in terms of noise level, blur and shooting conditions (lighting, inpainting, registration, occlusion, low resolution, etc.) [56,57,58,59,60,61] in order to automatically adjust the parameters necessary for an optimal use of the proposed DCSR algorithm.

The definitions of the acronyms used in this work are given in Table 4.

## Figures and Tables

**Figure 1 sensors-22-02199-f001:**
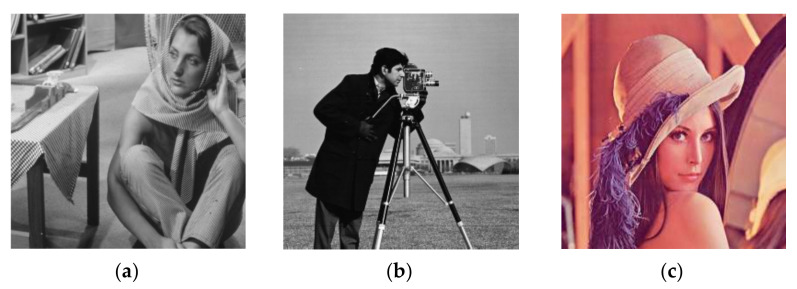
Test images. (**a**) Barbara, (**b**) Cameraman and (**c**) Lena.

**Figure 2 sensors-22-02199-f002:**
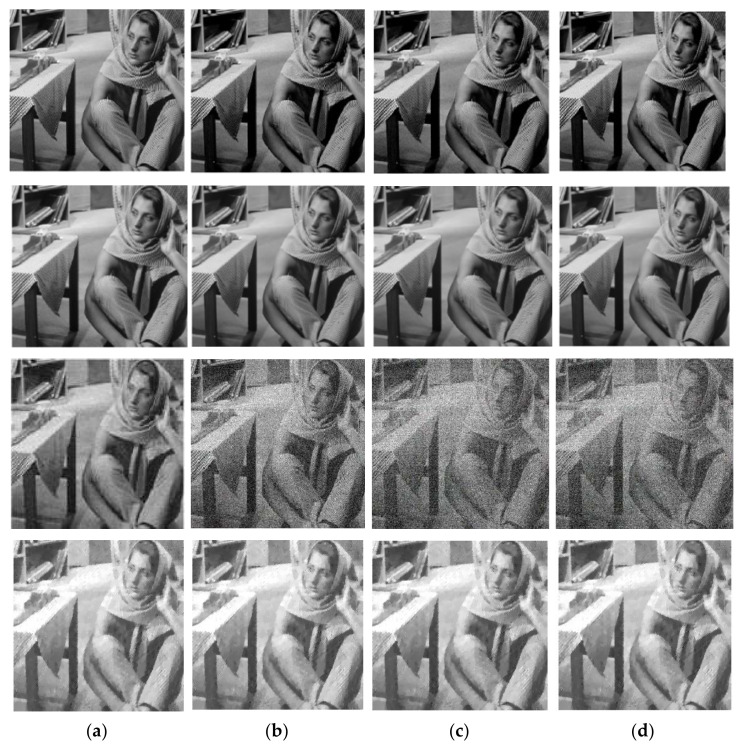
Visual comparison of the reconstruction quality for different noise levels. (**a**) σ=20, (**b**) σ=50, (**c**) σ=60 and (**d**) σ=80. The algorithms used are DCSR, GSR, wavelet denoising and NESTA: the denoised images are arranged in rows 1, 2, 3 and 4, respectively.

**Figure 3 sensors-22-02199-f003:**
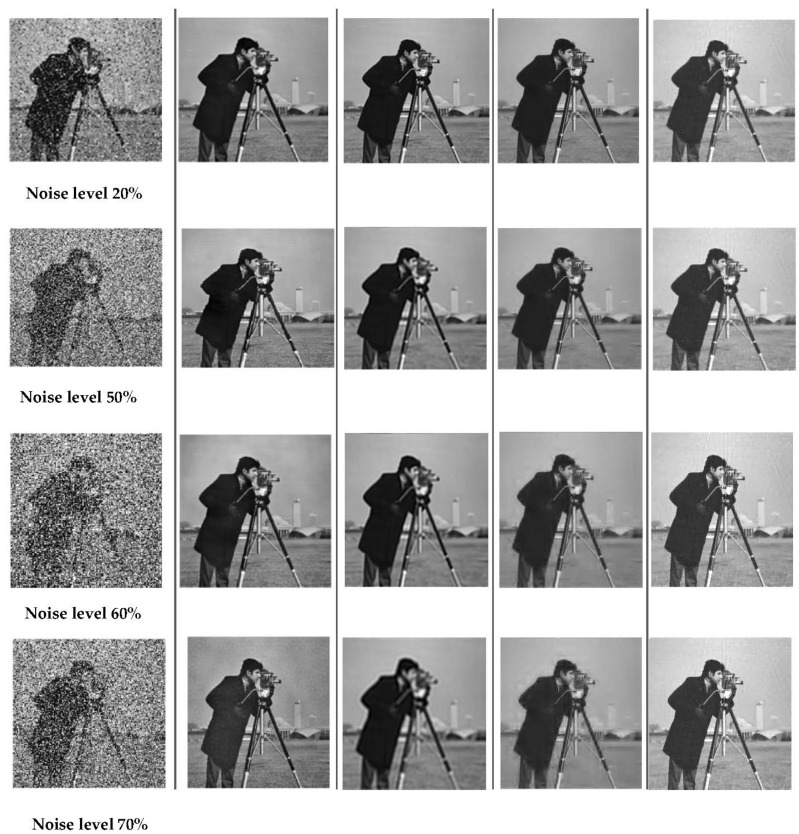
Visual comparison of the reconstruction quality for different noise levels in the case of salt and pepper (see column 1 located on the left side). The algorithms used are DCSR, GSR, wavelet denoising and NESTA: the denoised images are arranged in columns 2, 3, 4 and 5, respectively.

**Figure 4 sensors-22-02199-f004:**
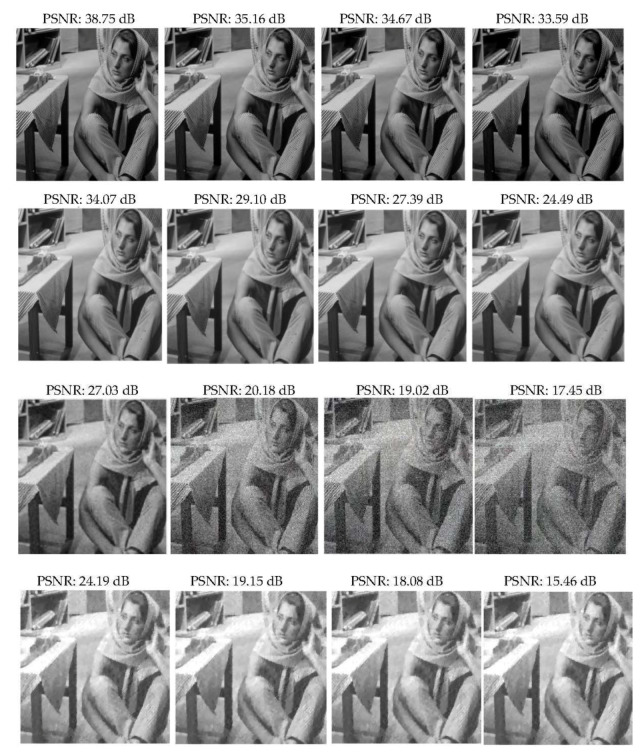
PSNR for different noise levels using the proposed DCSR, GSR algorithm, wavelet denoising and NESTA algorithm, arranged in rows 1, 2, 3 and 4, respectively. From left to right σ= 20, 50, 60 and 80, respectively.

**Figure 5 sensors-22-02199-f005:**
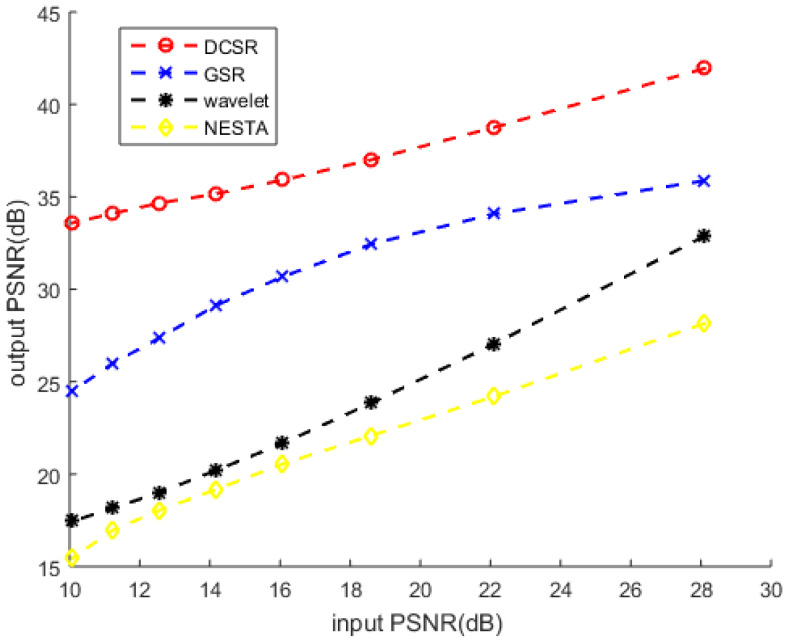
Grayscale image corrupted by additive white Gaussian noise (AWGN): performance analysis of four denoising methods.

**Figure 6 sensors-22-02199-f006:**
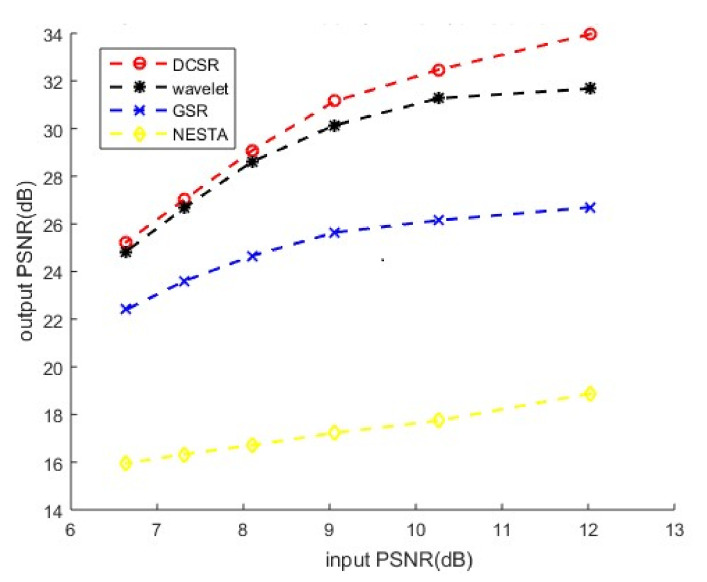
Grayscale image corrupted by salt and pepper noise: performance analysis of four denoising methods.

**Figure 7 sensors-22-02199-f007:**
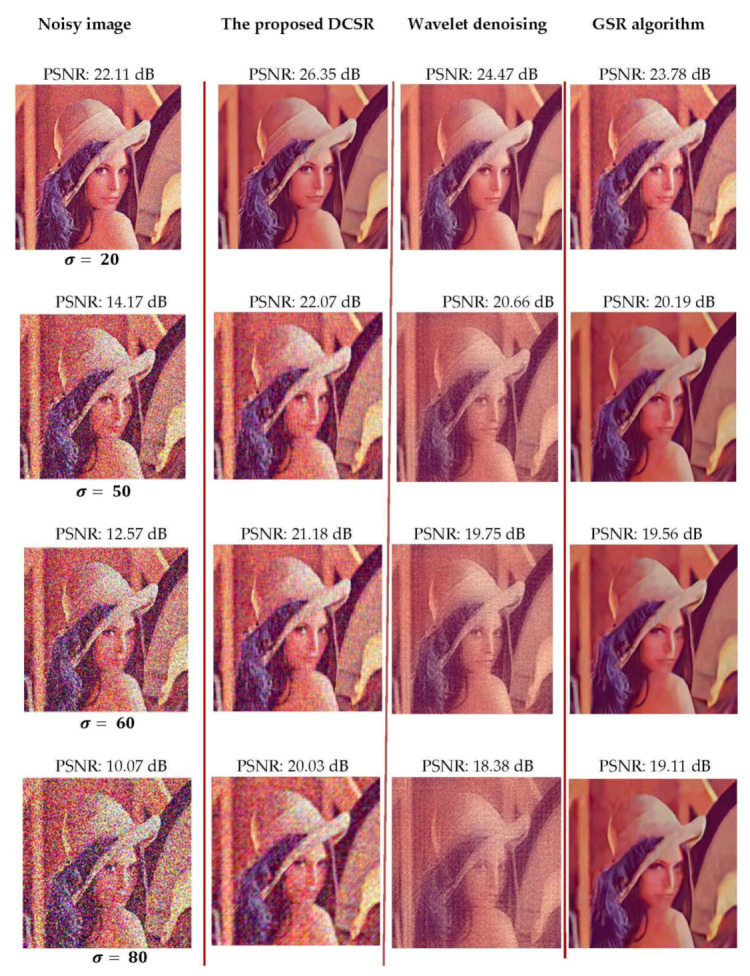
Restoration results of the AWGN-corrupted Lena image for different values of σ (see column 1 located on the left side of the figure). The algorithms used, the proposed DCSR, wavelet denoising and GSR, are arranged in columns 2, 3 and 4, respectively.

**Figure 8 sensors-22-02199-f008:**
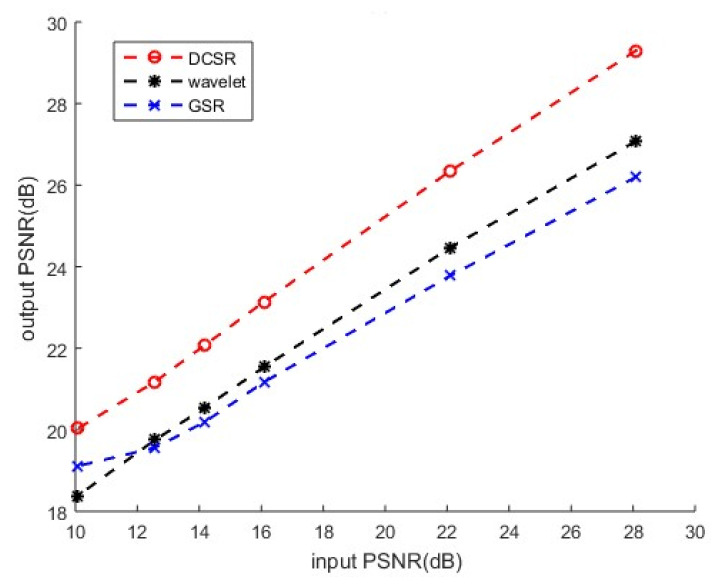
Color image corrupted by AWGN: performance analysis of three denoising methods.

**Figure 9 sensors-22-02199-f009:**
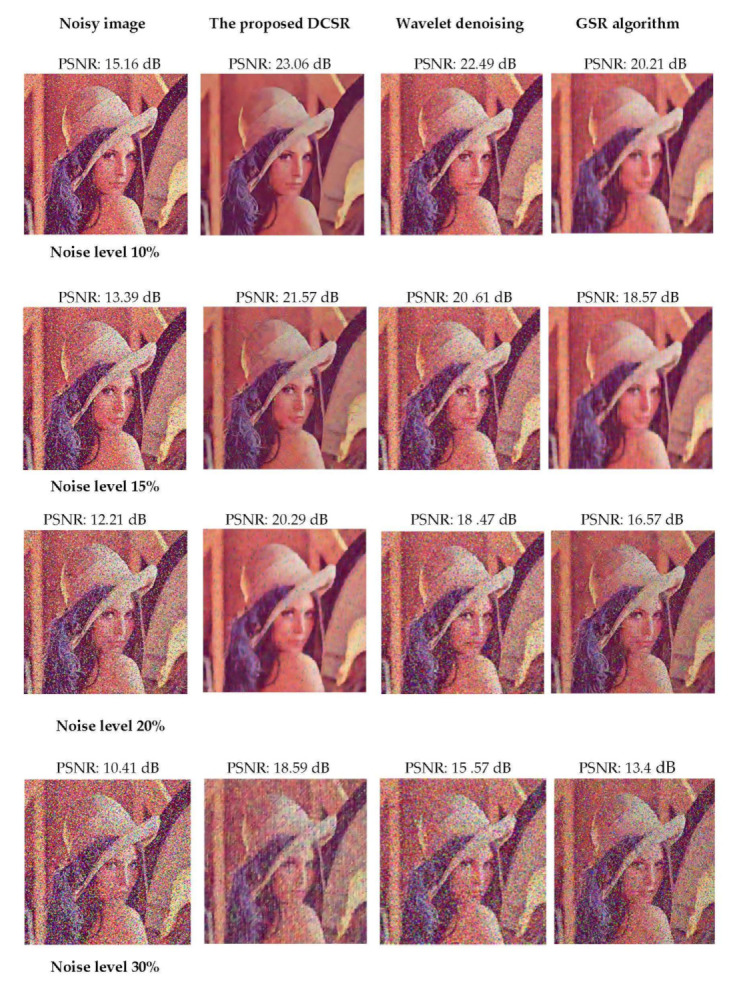
Restoration results of the salt and pepper noise-corrupted Lena image for different values of σ (see column 1 located on the left side of the figure). The algorithms used, the proposed DCSR, wavelet denoising and GSR, are arranged in columns 2, 3 and 4, respectively.

**Figure 10 sensors-22-02199-f010:**
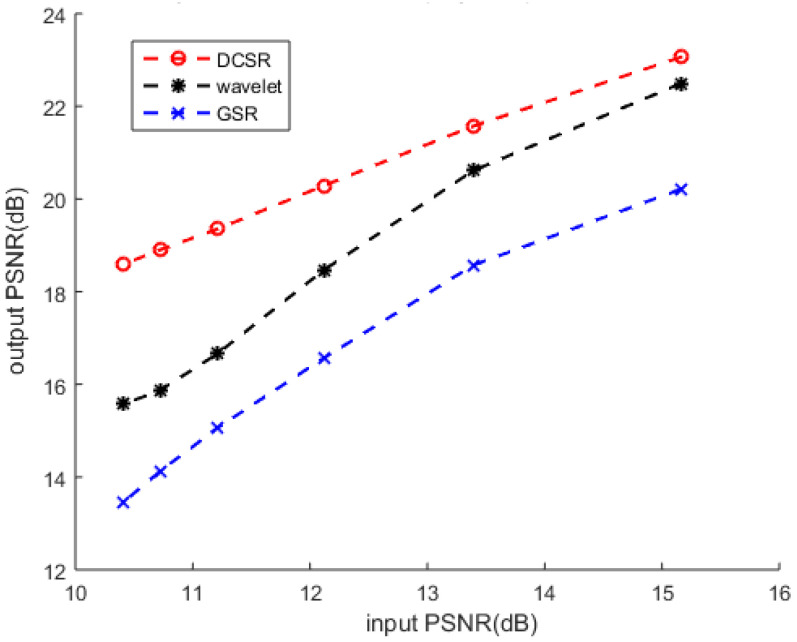
Color image corrupted by salt and pepper noise: performance analysis of three denoising methods.

**Figure 11 sensors-22-02199-f011:**
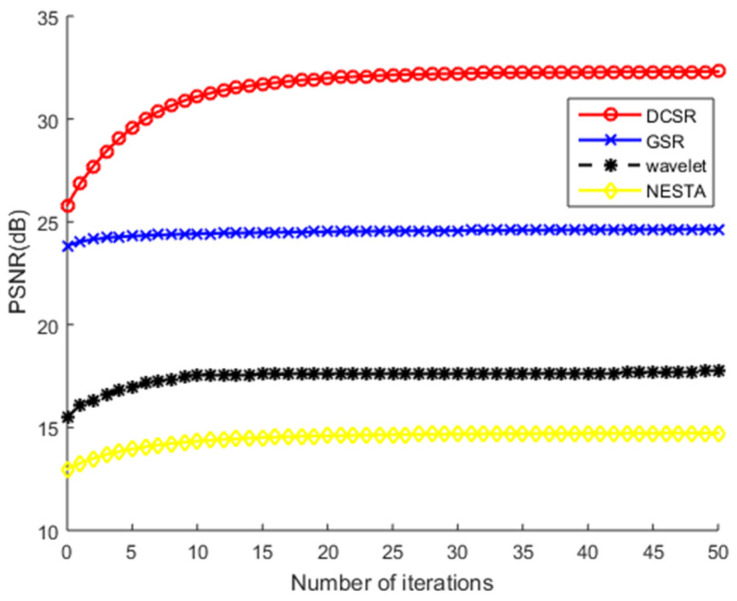
PSNR values of Barbara’s grayscale image recovered by four competing methods as a function of the number of iterations. The test image is corrupted by AWGN.

**Figure 12 sensors-22-02199-f012:**
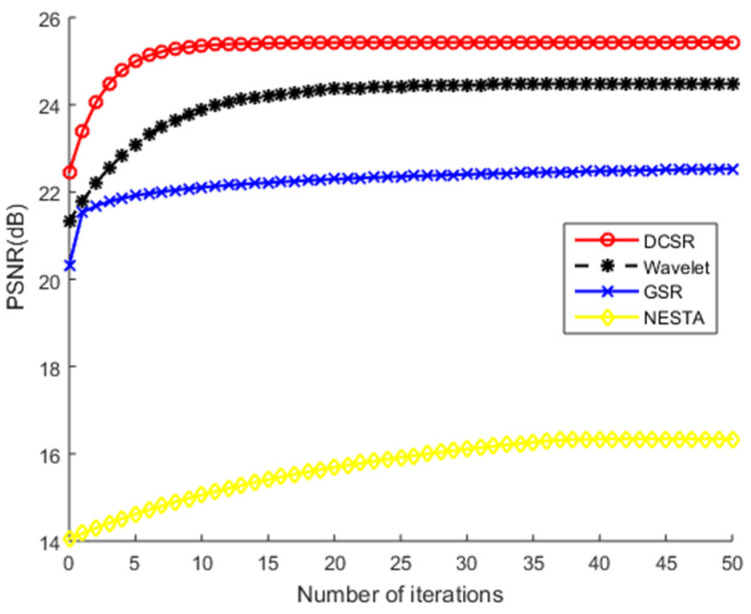
PSNR values of the Cameraman grayscale image recovered by four competing methods as a function of the number of iterations. The test image is corrupted by salt and pepper noise.

**Figure 13 sensors-22-02199-f013:**
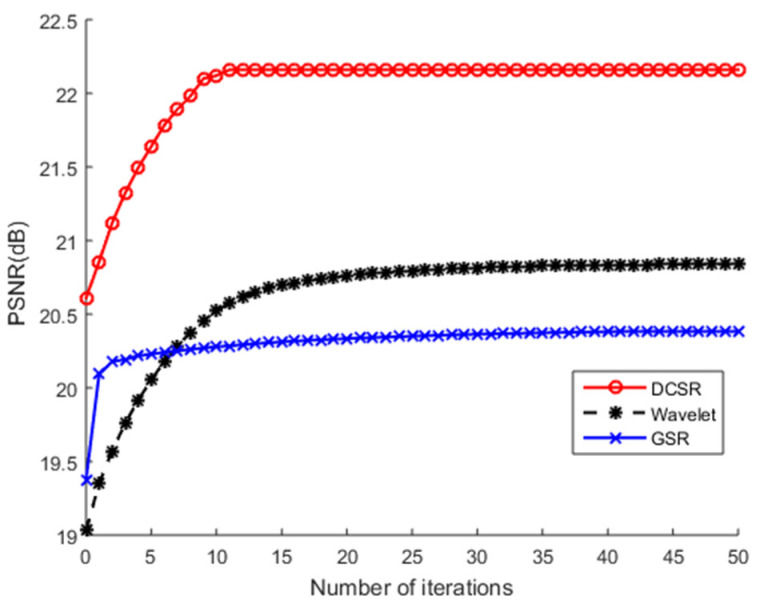
PSNR values of the recovered Lena color image by the competing methods vs. iteration number. The test image is corrupted by AWGN.

**Figure 14 sensors-22-02199-f014:**
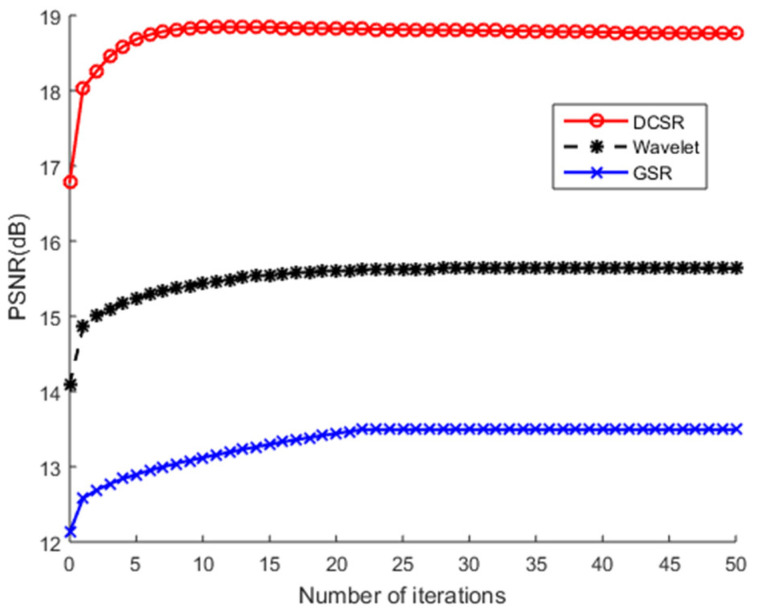
PSNR values of the recovered Lena color image by the competing methods vs. iteration number. The test image is corrupted by salt and pepper noise.

**Table 1 sensors-22-02199-t001:** Quality metric results on different algorithms for different values of noise levels.

Method	Noise level	PSNR	SSIM
Ours	20%	**33.97**	**0.99**
	50%	**30.08**	**0.95**
	60%	**27.03**	**0.93**
	70%	**25.24**	**0.92**
GSR algorithm	20%	26.70	0.80
	50%	24.66	0.78
	60%	23.61	0.74
	70%	22.40	0.69
Wavelet denoising	20%	31.68	0.89
	50%	28.61	0.86
	60%	26.69	0.79
	70%	24.84	0.74
NESTA algorithm	20%	18.88	0.39
	50%	16.71	0.38
	60%	16.33	0.37
	70%	16.03	0.36

**Table 2 sensors-22-02199-t002:** Computational complexity.

Algorithms	Computational Complexity
TV	$O\left(N\right)\;$ [46]
Wavelet denoising	$O\left(N\log_{2}N\right)\;$ [49]
NESTA	$O\left(N+N\log_{2}N\right)\;$ [48]
GSR	$O\left(N\;\left(n\;m^{2}+T_{s}\right)\right)$ [47]
Our algorithm DCSR	$O\left(N\left(n\;m^{2}+T_{s}\right)+N\right)$

**Table 3 sensors-22-02199-t003:** Ablation analysis.

Method	σ	PSNR	SSIM
CS	20	22.11	0.53
	50	20.35	0.43
	60	18.59	0.37
CS+TV	20	22.34	0.54
	50	21.74	0.52
	60	20.67	0.44
CS+NLS	20	29.45	0.89
	50	28.57	0.78
	60	27.97	0.74
CS+TV+NLS (DCSR)	20	**38.75**	**0.99**
	50	**35.16**	**0.97**
	60	**34.67**	**0.96**

**Table 4 sensors-22-02199-t004:** Definition of acronyms.

Acronyms	Description
CS	Compressed Sensing
DCSR	Denoising Compressed Sensing by Regularizations terms
PSNR	Peak Signal-to-Noise Ratio
SSIM	Structural Similarity
TV	Total Variation
NLS	Nonlocal self-Similarity
DWT	Discrete Wavelet Transform
NESTA	Nesterov’s algorithm
GSR	Group-based Sparse Representation
AWGN	Additive White Gaussian Noise
MSE	Mean Square Error

## Data Availability

Not applicable.

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
