# Peer review of "Image Denoising Using a Compressive Sensing Approach Based on Regularization Constraints"

_sensors, 2022, doi:10.3390/s22062199_

Round 1

Reviewer 1 Report

Reviewer’s Report on the manuscript entitled:

Image denoising using a compressive sensing approach based on regularization constraints

The authors proposed a compressed sensing reconstruction algorithm for image denoising that combines the total variation regularization and the non-local self-similarity constraint. Image denoising is a widely studied topic, and the authors should improve the literature review. In my view, the method and results are interesting. However, the presentation can be further improved. Below please see my comments.

Line 27. Please define the acronyms. Please note that all the abbreviations must be defined the first time they appear in the Abstract and the body of the manuscript. Please be consistent with their style and add an acronym table at the end of the manuscript listing all the acronyms used in the manuscript.

Line 43-55. Please avoid mathematical formulas in the Introduction. These can go to the method section.

Lines 56-62. Please also add the following articles for regularization and random noise attenuation with a brief description for each:

A robust spectral analysis technique for sparse signal and image regularization:

https://doi.org/10.1007/s11600-019-00320-3

Image denoising using a deep generative network:

https://doi.org/10.3390/app11114803

Image denoising via low-rank tensor approximation:

https://doi.org/10.3390/rs12081278

Equation (13). Is the sum over i? Please below the sum symbol say i=1.

Lines 197, 199, etc. Please use "Equation" instead of "problem", so here say “Equation (19)”. Please check and correct everywhere else.

Please discuss the artifacts observed above the head of the camera and on the top-right corner of the images using NESTA in the fifth column of Figure 6.

The first column in Figure 6. The noise is too high that the cameraman is not even visible let alone the background! I don’t think any method can recover this! Please check and correct. The illustration of the first column in Figure 13 is more realistic.

Figures 2,3,4,5 can go all to one figure just like Figure 6 (the cameraman).

Figures 7,8,9,10 can go to one figure just like Figure 13.

Figure 13. The sizes of some of the images in the last column are not the same. Please correct. All the images should have the same size.

Figures 18, 20. What happened to NESTA in yellow? Please add that too.

A discussion section is required before the Conclusions. Please note that the results section should have just the numerical results and illustrations and short descriptions, but any comments on the results should be in the discussion. Please discuss the problem and your results in the light of other similar methods, and mention its advantages, limitations, etc. The use of l2 norm is for smoothing purposes while l1 tends to keep the edge information. I think your proposed method is a mix of both. This also is true for the use of wavelets (e.g., Morlet for smoothing and Haar for keeping edge information). Some detailed discussion like this is also needed here. Please keep in mind that mathematical formulas can distract the general readers from understanding the concept. So, please use plain language without any formulas in the Discussion section.

Thank you for your contribution

Regards,

Reviewer 2 Report

This paper proposes a method to preserve the texture of the image and tend to smooth out its details. The proposed algorithm denoises compressed sensing by regularization terms, which can perform image reconstruction and denoising. Generally speaking, reconstruction techniques are often unable to preserve the texture of an image and tend to smooth out its details. The proposed method is simple and effective, with good improvement in both denoising efficiency and visual quality. In my opinion, the article has relatively strong originality, and meets the requirements of publication. I recommend accepting it but with some modifications.

  1. Make sure your conclusions reflect on the strengths and weaknesses of your work, how others in the field can benefit from it and thoroughly discus future work. The conclusions should be different in content from the abstract, and be rather comprehensive too.
  2. In the abstract and main text, "regularization terms" and "regularization terms" appear. Is there any difference between the two?
  3. The authors ignore some relevant papers. For example, image processing method that has been published in 2021 please see (“A Novel Fast Single Image Dehazing Algorithm Based on Artificial Multiexposure Image Fusion,” IEEE Transactions on Instrumentation and Measurement, vol.70). The authors should compare their methods with it carefully.
  4. In Section 4, too many formulas and too few explanations make the logic of the method not shown. The authors are strongly encouraged to add necessary descriptions to explain the authors' reasons for doing so to improve method coherence.
  5. For clarity, the authors should verify the performance of the proposed method. That is, the authors need to provide the results of ablation experiments with or without the proposed method.
  6. Most of references are a little bit out of date. In particular, most of the references compared in the experiments are from 2019 and before. Please discuss recently released solutions, especially those released in 2021 and 2020.
  7. Missing articles and other incorrect English language constructs are distracting and make interpretation difficult. The authors are advised to correct the linguistic errors in the edited text.

Round 2

Reviewer 1 Report

I would like to thank the authors for addressing my comments. The manuscript is significantly improved including the quality of the figures. Please see a few minor suggestions below:

Lines 15 and 18. Please replace CS with compressed sensing.

In the the highlighted line after Equation (11): please remove the hat symbol after u.

You copied the same words as in Lines 45-47 to put in lines 599-601. Please rephrase lines 599-601 to avoid repetition. 

Lines 617-618, Lines 666-667, Lines 673-674. A paragraph should have at least three sentences. So, please either add more sentences there or merge them with the previous or following paragraphs.

Line 48. Please say "From Figure 4...".

Line 632. Style issue. Please say "Figures 11-14"

Please carefully proofread the article before publication.

Thank you for your contribution

Regards,

Reviewer 2 Report

All my concerned problems are revised.

Author Response

Thank you for your help.

Round 3

Reviewer 1 Report

Thank you for addressing my remaining comments. I have no further comments.

Thank you for your contribution

Regards,